# DNA-Dependent Protein Kinase Catalytic Subunit: The Sensor for DNA Double-Strand Breaks Structurally and Functionally Related to Ataxia Telangiectasia Mutated

**DOI:** 10.3390/genes12081143

**Published:** 2021-07-27

**Authors:** Yoshihisa Matsumoto, Anie Day D. C. Asa, Chaity Modak, Mikio Shimada

**Affiliations:** Laboratory for Zero-Carbon Energy, Institute of Innovative Research, Tokyo Institute of Technology, Tokyo 152-8550, Japan; asa.a.aa@m.titech.ac.jp (A.D.D.C.A.); modak.c.aa@m.titech.ac.jp (C.M.); mshimada@zc.iir.titech.ac.jp (M.S.)

**Keywords:** DNA-dependent protein kinase (DNA-PK), DNA-dependent protein kinase catalytic subunit (DNA-PKcs), Ku, Ataxia–telangiectasia mutated (ATM), protein kinase, phosphatidylinositol 3-kinase-like kinase (PIKK), DNA double-stranded break (DSB), DNA damage response

## Abstract

The DNA-dependent protein kinase (DNA-PK) is composed of a DNA-dependent protein kinase catalytic subunit (DNA-PKcs) and Ku70/Ku80 heterodimer. DNA-PK is thought to act as the “sensor” for DNA double-stranded breaks (DSB), which are considered the most deleterious type of DNA damage. In particular, DNA-PKcs and Ku are shown to be essential for DSB repair through nonhomologous end joining (NHEJ). The phenotypes of animals and human individuals with defective DNA-PKcs or Ku functions indicate their essential roles in these developments, especially in neuronal and immune systems. DNA-PKcs are structurally related to Ataxia–telangiectasia mutated (ATM), which is also implicated in the cellular responses to DSBs. DNA-PKcs and ATM constitute the phosphatidylinositol 3-kinase-like kinases (PIKKs) family with several other molecules. Here, we review the accumulated knowledge on the functions of DNA-PKcs, mainly based on the phenotypes of DNA-PKcs-deficient cells in animals and human individuals, and also discuss its relationship with ATM in the maintenance of genomic stability.

## 1. DNA-PKcs as a Family Member of ATM

### 1.1. DNA-PK and DNA-PKcs

The first observation of DNA-dependent protein kinase (DNA-PK) activity dates back to 1985. Walker et al. showed that double-stranded DNA (dsDNA), but not single-stranded DNA or RNA, induced the phosphorylation of multiple proteins, including Heat-shock protein 90 (Hsp90), in a wide variety of eukaryotic cells of human, rabbit, frog and sea urchin origin [1]. Carter et al. and Lees-Miller et al. purified this enzymatic activity from the nuclear extract of human cervical carcinoma HeLa cells and found it to be associated with a 300–350-kDa polypeptide, which is now known as the DNA-dependent protein kinase catalytic subunit (DNA-PKcs) [2,3].

Lees-Miller et al. noticed that the two polypeptides that can be phosphorylated by DNA-PK were copurified in later steps of purification and identified them as the Ku70/Ku80 heterodimer (hereafter referred to as Ku, simply) [3]. Dynan and colleagues purified a transcription template-associated kinase, which phosphorylates heptapeptide repeats in the C-terminal domain of the largest subunit of RNA polymerase II, and showed that it consists of two components, i.e., a 300–350-kDa polypeptide and Ku [4]. Jackson and colleagues showed that DNA-PK binds to and requires the end of dsDNA for activity [5]. They demonstrated that Ku is an essential component of DNA-PK, mediating the binding of DNA-PKcs to the dsDNA end [5].

The Ku protein was initially identified by Mimori et al. as the antigen of the autoantibody in a patient of an autoimmune disease, polymyositis–scleroderma overlap [6]. Subsequent studies showed that the Ku protein consists of two polypeptides of 70 kDa and 80 kDa, which are termed Ku70 and Ku80 (also termed Ku86), respectively [7], and binds to the end of double-stranded DNA without a particular preference in the nucleotide sequences [8]. Later, X-ray crystallography revealed that Ku70 and Ku80 form a ring-shaped structure that can encircle DNA, giving a plausible explanation for how Ku binds specifically to the DNA end [9].

The nature of DNA-PK suggests its potential as a “sensor” for DSB. In 1995, Blunt et al. [10], Kirchgessner et al. [11] and Peterson et al. [12] showed that DNA-PKcs are inactivated in severe combined immunodeficiency (SCID) mice. SCID mice were initially established by Bosma et al. in 1983 [13]. SCID mice are defective in the V(D)J recombination of immunoglobulin and T-cell receptor genes (see below and Appendix A) and, as a result, lack B-cell- and T-cell-mediated immunity [14,15]. Cells from SCID mice are sensitive to IR due to a reduced ability in DNA double-stranded break (DSB) repair [16]. In addition, Lees-Miller et al. reported the absence of DNA-PKcs in M059J, which was established from a human malignant glioma specimen, showing radiosensitivity and defective DSB repair [17]. In the year before, Taccioli et al. [18] and Smider et al. [19] showed that Ku80 is inactivated in IR-sensitive and V(D)J recombination-defective hamster cells, such as *xrs*-5, XR-V15B and *sxi*-3. These discoveries indicated that DNA-PKcs and Ku are essential for DSB repair and V(D)J recombination. This historical perspectives have been reviewed more in detail elsewhere [20].

### 1.2. DNA-PKcs and ATM

The molecular cloning of cDNA-encoding DNA-PKcs was achieved in 1995 and revealed its extremely large size, i.e., 4128 amino acids, which was even larger than expected before [21]. Another surprise was its similarity to Ataxia–telangiectasia mutated (ATM), the gene responsible for ataxia–telangiectasia (AT), which was reported shortly before (Figure 1a) [22]. AT is an autosomal recessive human genetic disorder that shows pleiotropic effects, including defective locomotive regulation (ataxia), the enlargement of capillaries (telangiectasia), immunodeficiency, infertility and an increased risk of cancer. Cells from AT patients exhibit a high sensitivity to IR, showing “radioresistant DNA synthesis”, which is an indicative of defective cell cycle checkpoints (especially in the G1/S- and S-phase checkpoints). The primary structure of ATM showed a similarity to phosphatidylinositol 3-kinase [22]. It raised the question of the role of lipid phosphorylation in cell cycle checkpoints, if any. Simultaneously, several genes implicated in cell cycle checkpoint and/or telomere maintenance, i.e., Mei-41 [23] in *Drosophila melanogaster* and TEL1 [24,25] and MEC1 [26] in *Saccharomyces cerevisiae*, were found to be structurally related to ATM. Subsequently, ATM- and Rad3-related ATR (also termed FRAP-associated protein-1, FRP1) was found in humans [27,28]. ATM and ATR are human orthologs of *Saccharomyces cerevisiae* TEL1 and MEC1, respectively.

These molecules comprise a family of protein kinases termed phosphatidylinositol 3-kinase-related kinases (PIKKs). In addition to DNA-PKcs, ATM and ATR, three other members of the PIKK family have been identified in humans (Figure 1a). Suppressor of morphological defects on genitalia-1 (SMG-1) is involved in the nonsense-mediated decay of mRNA [29,30]. Mammalian Target of rapamycin (mTOR, also termed FKBP12-rapamycin-associated protein, FRAP, and Rapamycin and FKBP12-target, RAFT) regulates cell growth and survival, sensing amino acids and growth factors [31,32]. Transformation/transcription domain-associated protein (TRRAP) is involved in chromatin modification and remodeling and, interestingly, lacks catalytic activity [33]. In addition to the kinase catalytic domain, these molecules show structural similarity in the FAT (FRAP, ATM and TRRAP); PRD (PIKK-regulatory domain) and FATC (FAT C-terminal) domains. They also have less-conserved HEAT (Huntingtin, Elongation factor 3, Protein phosphatase 2A and TOR1) repeats.

Without the abundant knowledge on DNA-PKcs, these molecules might have been assumed to be lipid kinases rather than protein kinases. However, since DNA-PK is known as an unambiguous protein kinase, these molecules were presumed to be protein kinases, probably sharing similar properties with DNA-PK. It was known that DNA-PK preferentially phosphorylates serines or threonines, followed by glutamines (SQ/TQ motif), including Ser15 of p53 [34,35]. Subsequently, it was shown that ATM and ATR were capable of phosphorylating Ser15 of p53 [36,37]. Thus, SQ/TQ motifs are considered the consensus sequences for ATM and ATR, as well as DNA-PKcs. Furthermore, it was shown that ATM rather than DNA-PKcs are mainly responsible for the phosphorylation of Ser15 of p53 in cellulo in response to DNA damage [36,37]. However, these studies also noted a difference in the biochemical property: where DNA-PKcs require Mg^2+^ for activity, ATM and ATR require Mn^2+^ [36,37].

While DNA-PK requires Ku for DNA binding and activation, ATM requires the Mre11-Rad50-Nbs1 (MRN) complex [38,39]. ATR binds to ssDNA via Replication protein A (RPA) and ATR-interacting protein (ATRIP) [40]. In the C-termini of Ku80, Nbs1 and ATRIP, there are similar amino acid sequences (EEX_3-4_DDL, where X represents any amino acid), which are required for the recruitment of DNA-PKcs, ATM and ATR, respectively [41].

DNA-PKcs and ATM were implicated in the DNA damage response, especially for DSBs. Their similarities in structure and properties raise the question of what the overlapping function(s) and nonoverlapping function(s) between the two molecules is/are. As described above, the defects in DSB repair and V(D)J recombination manifested in DNA-PKcs-deficient cells. On the other hand, the defects in cell cycle checkpoints were manifested in cells from AT patients. Hence, it seems that DNA-PK is mainly involved in DSB repair and V(D)J recombination, whereas ATM is mainly involved in cell cycle checkpoints.

## 2. Function and Role of DNA-PKcs

### 2.1. DNA-PKcs in Nonhomologous End Joining

There are two pathways to repair DSBs, which are evolutionally conserved throughout eukaryotes: homologous recombination (HR) and nonhomologous end joining (NHEJ) [42]. HR uses a homologous or identical sequence as the template to reconstitute the DNA sequence around DSB. On the other hand, NHEJ joins two DNA ends in the close vicinity with minimally required modifications (termed processing). Hence, NHEJ may occasionally incur errors, such as nucleotide deletions or insertions at the junction, or ligation with an incorrect partner, leading to chromosomal aberrations such as deletions, inversions, or translocations. Hence, HR is generally considered more accurate than NHEJ. However, HR requires the sister chromatids in vertebrates and is, therefore, restricted in the late S and G2 phases, whereas the majority of cells are in the G0 and G1 phases. Even in the G2 phase, approximately 80% of DSBs are thought to be repaired via NHEJ in human cells [42]. Moreover, as only a small portion of the genome (~1% or less) in vertebrates encodes protein [43], the insertion or deletion of a small number of nucleotides can be tolerated. Thus, NHEJ is thought to have prominent importance, especially in vertebrates, including humans. In addition, NHEJ is also involved in V(D)J recombination [42].

Throughout the 1970s and 1980s, a number of ionizing radiation (IR)-sensitive mutants were isolated from rodent cells. These were classified into 11 complementation groups, and the gene complementing the IR sensitivity of each group was named XRCC1–XRCC11, respectively, where XRCC stands for “X-ray cross-complementing” [44]. The cells that were classified into groups 4, 5 and 7 showed similar characteristics, i.e., the defects in rejoining of IR-induced DSBs and V(D)J recombination [44]. DNA-PKcs and Ku80 correspond to XRCC7 and XRCC5, respectively. XRCC4, which is absent in IR-sensitive and V(D)J recombination defective cells, i.e., hamster XR-1, was identified in 1995 by Li et al. [45]. Subsequently, XRCC4 was shown to be associated with DNA ligase IV (LIG4) and essential for the function and stability of LIG4 [46,47]. Mutations in the LIG4 gene were found in an X-ray-sensitive mutant derived from murine mammary carcinoma cells [48] and in a cell line established from a radiosensitive leukemia patient [49]. Based on these, LIG4 is thought to catalyze the ligation of two DNA ends to complete NHEJ. In 2001, Artemis was found as the gene mutated in human radiosensitive-SCID (RS-SCID) patients [50]. Subsequently, Artemis was shown to be capable of opening a hairpin structure, which appears at the coding ends (Appendix A), and processing overhangs, which result from the hairpin opening [51]. Moreover, DNA-PKcs were shown to form a complex with Artemis in vitro and in vivo and, also, to regulate the above activities of Artemis through phosphorylation in vitro [51]. In 2006, XRCC4-like factor (XLF), which is also known as Cernunnos, was identified as an XRCC4-interacting protein in a two-hybrid screening and, simultaneously, as the gene mutated in human RS-SCID patients with microcephaly [52,53]. XLF also showed a similarity to XRCC4 in the predicted 3D structure and is thought to be a paralog of XRCC4 [53,54]. XLF was shown to stimulate LIG4 activity for mismatched or noncompatible ends [55,56] and, also, to form filaments with XRCC4, which may have some role in aligning or bridging two DNA ends [57,58,59,60]. In 2015, Paralog of XRCC4 and XLF (PAXX, also termed XLS for XRCC4-like small molecule) was identified simultaneously by three groups [61,62,63]. PAXX was shown to stabilize the binding of Ku to DSBs and the assembly of the NHEJ complex therein [61,62,63].

Figure 2 illustrates a simplified diagram of DSB repair through NHEJ. It should be noted that NHEJ is much more sophisticated, involving many other processing enzymes and chromatin remodelers [20,42], but DNA-PKcs, Ku and the other above-mentioned “core” NHEJ factors are illustrated here. DNA-PKcs are thought to be involved in the recognition of DSBs with Ku and orchestrate the subsequent steps of NHEJ. The kinase activity of DNA-PKcs is considered essential, because it was shown that the catalytically inactive forms of DNA-PKcs were not sufficient to restore the normal radiosensitivity and V(D)J recombination ability in DNA-PKcs-deficient cells [64,65]. However, the target(s) and role(s) of phosphorylation in NHEJ remain to be clarified, as discussed elsewhere [20].

### 2.2. Cells Deficient for DNA-PKcs: Role in DSB Repair and V(D)J Recombination

Table 1 summarizes the DNA-PKcs-deficient cells and their characteristics. In addition to murine scid and human M059J, several mouse and hamster cell lines, which were isolated as IR-sensitive mutants, turned out to be deficient in DNA-PKcs. Furthermore, several DNA-PKcs-deficient cell lines were engineered by gene targeting or, more recently, genome editing. Almost all of them showed an increased sensitivity toward ionizing radiation and drugs, like etoposide and bleomycin, which induce DSBs. A reduced ability in DSB repair of these cells was demonstrated by pulse-field gel electrophoresis, mainly in the 1990s, and, thereafter, by the foci of DSB markers like γ-H2AX and 53BP1. These results indicate the essential roles of DNA-PKcs in DSB repair.

Table 2 summarizes the V(D)J recombination ability of DNA-PKcs-deficient cells, measured using extrachromosomal substrates with an exogenous expression of Rag1 and Rag2 (See Appendix A). Coding joint formation is reported to be greatly affected in DNA-PKcs-deficient cells, except for XR-C2. However, there is some controversy in the signal joint formation. Some study reported that the frequency of the signal joint was only slightly decreased or not affected at all, whereas others reported significant reductions. Regarding the fidelity, there have been studies reporting only a modest reduction and those reporting profound or complete impairment. The exact reason for this controversy is unknown, but it may have arisen from the differences in DNA-PKcs mutations and/or in cellular backgrounds. In any case, the cording joint formation is more impaired than the signal joint formation in DNA-PKcs-deficient cells. In Ku80- or XRCC4-deficient cells, both the signal joint formation and the coding joint formation are significantly impaired (Table 2). Considering this, the requirement for DNA-PKcs in signal joint formation seems to be less than that for Ku80 or XRCC4. Coding joint formation is thought to be more difficult than signal joint formation, as the former requires opening the hairpin and processing the overhang. Hence, DNA-PKcs might be dispensable for the ligation of “clean” ends, such as blunt ends, but be required for the ligation of “difficult” or “dirty” ends, such as hairpin ends, which need end processing.

### 2.3. Animals Deficient for DNA-PKcs: Role in Development

Table 3 summarizes the DNA-PKcs-deficient animals and their characteristics. The animals show the SCID phenotype, and the cells show increased sensitivity to IR and defective V(D)J recombinations.

Mice with *scid* mutations show the absence of mature T and B lymphocytes [13]. The *scid* mice are highly susceptible to infection by bacteria, viruses and fungi because of an inability to generate an antigen-specific immune response. In addition, *scid* mice lack transplant rejection and are used for xenografts. However, *scid* mice are termed “leaky”, as they can produce some immunoglobulins and T lymphocytes at increased ages [83]. Murine *scid* mutations lead to a lack of ~2% of the C-terminal region, although the protein expression is greatly diminished, and the protein kinase activity is undetectable. Where the coding joint formation is almost completely abrogated, the signal joint formation remains, at least partially. This has raised the possibility that DNA-PKcs in *scid* is not functionally null. As opposed to this, however, DNA-PKcs knockout (DNA-PKcs^−/−^) mice, which were generated by three groups independently, were capable of signal joint formation [84,85,86]. In addition, mice with slip mutations, which are generated incidentally by the insertion of a transgene to the DNA-PKcs gene and thought to be functionally null, also show a ~10% ability of signal joint formation without a decrease in fidelity [87,88]. These lines of evidence indicate that DNA-PKcs are not absolutely required for signal joint formation.

Knockin mice lacking kinase activity (D3922A substitution in the kinase domain, hereafter referred to as KD for kinase dead) and that lack three autophosphorylation sites (T2605/2634/2643A, hereafter referred to as 3A) were generated [89,90]. In DNA-PKcs^KD/KD^ mice, the abundance of DNA-PKcs appeared normal, but its kinase activity was undetectable [89]. DNA-PKcs^KD/KD^ mice showed late embryonic lethality, dying before embryonic day 14.5 (E14.5) [89]. In the brains of DNA-PKcs^KD/KD^ mice, extensive apoptosis was observed at a level similar to LIG4^−/−^ or XRCC4^−/−^ mice (see below) [89]. In DNA-PKcs^3A/3A^ mice, the expression of DNA-PKcs and its kinase activity were normal [90]. Although DNA-PKcs^3A/3A^ mice were born normally in terms of ratio and size, they become smaller within 2 to 3 weeks of age and died shortly after birth [90]. Congenital bone marrow failure and loss of hematopoietic stem cells were observed in DNA-PKcs^3A/3A^ mice [90]. The lifespans of DNA-PKcs^3A/3A^ mice were extended in p53^+/−^ and p53^−/−^ backgrounds with a concomitant alleviation in lymphocyte development defects [90]. This observation suggests that a shortened lifespan and lymphocyte development defects are at least partially due to p53-mediated DNA damage responses, including apoptosis. Cells from DNA-PKcs^3A/3A^ mice show an elevated sensitivity to DNA crosslinking agents, as well as IR [90]. These characteristics of DNA-PKcs^3A/3A^ mice were similar to those of Fanconi’s anemia. More severe phenotypes of DNA-PKcs^KD/KD^ mice and DNA-PKcs^3A/3A^ mice than that of DNA-PKcs^−/−^ mice might be related to the mechanisms of the regulation of DNA-PKcs through phosphorylation by itself and ATM, as we will discuss below.

It might be noted that mice of several strains, including C.B.17, from which *scid* mice were derived, have hypomorphic DNA-PKcs, although immunologically normal [91,92]. In the Balb/c strain, the expression of DNA-PKcs and DNA-PK kinase activity was 5–10% of the corresponding tissues and cells from the C57BL/6 strain [91,92,93]. There were two nucleotide substitutions in the DNA-PKcs gene and resultant amino acid substitutions in the protein in Balb/c compared to C57BL/6: c.C6418T, p.R2140C and c.A11530G and p.M3844V. The Balb/c mice were susceptible to breast cancer and thymic lymphoma and showed increased apoptosis in the thymuses after irradiation [92,93]. The fibroblasts from Balb/c mice showed a reduced DSB repair ability, which is an intermediate of C57BL6 and *scid* mice. In crossing experiments, the Balb/c allele was associated with an increased risk of thymic lymphoma and chromosome aberrations [92,93].

SCID in horse (Arabian foal) was found earlier than that in mice [94]. Soon after the finding in mice, SCID horses were shown to be deficient in DNA-PKcs [95,96]. Unlike the case of *scid* mice, SCID horses are not reported to be leaky and incapable of signal joint formation, as well as coding joint formation [95,96]. SCID animals was also found in dogs (Jack Russel Terriers) and shown to harbor mutations in DNA-PKcs [97,98]. SCID dogs showed intermediate activity in both the signal joint formation and the coding joint formation [98]. Thus, the requirement for DNA-PKcs in V(D)J recombination may differ among species. Meek et al. noted that it may be related to an abundance of DNA-PKcs expression and/or DNA-PK kinase activity; the kinase activity, as well as the requirement for DNA-PKcs in V(D)J recombination is the highest in horses and the lowest in mice.

DNA-PKcs^−/−^ rats, generated through Zinc-finger nuclease (ZFN)-mediated genome editing, showed SCID without leakiness. In addition, DNA-PKcs^−/^^−^ rats showed growth retardation, i.e., smaller body sizes than age-matched DNA-PKcs^+/+^ or DNA-PK^+/^^−^ rats [99], which was not noticed in mice, horses and dogs. In agreement with this, the fibroblasts from DNA-PKcs^−/−^ rats showed reduced proliferation and premature senescence [99]. A reduction in litter size was also noted in DNA-PKcs^−/−^ rats [99].

Besides mammals, DNA-PKcs^−/−^ animals were also generated in zebrafish through transcription activator-like effector nuclease (TALEN)-mediated genome editing [100,101]. DNA-PKcs^−/^^−^ zebrafish also showed the SCID phenotype and competency for xenograft experiments [100,101]. Growth retardation was noted at lower ages, i.e., up to 20 weeks, although it was not obvious thereafter [100].

Table 4 shows the comparison of the phenotypes of DNA-PKcs^−/−^ mice and those of other genes, showing similarities and dissimilarities. Ku80^−/−^ mice mostly show a complete absence of mature B and T lymphocytes and defects in signal joint formation and coding joint formation [102,103]. Ku70^−/−^ mice showed SCID with some leakiness; although mature B lymphocytes and serum immunoglobulins were absent, mature T lymphocytes were present, albeit reduced [104,105]. Mouse embryonic fibroblasts (MEF) from Ku80^−/−^ mice and Ku70^−/−^ mice were defective in both coding joint formation and signal joint formation in V(D)J recombination. The reason for the different impact of the loss of Ku80 and Ku70 on the T lymphocytes is not known. It may be noted that Ku70^−/−^ mice, but not Ku80^−/−^ mice, showed an increased frequency of thymic lymphoma [104,106]. Both Ku80^−/−^ mice and Ku70^−/−^ mice showed a reduction in body sizes, i.e., 40–60% of the control and litter sizes [102,103,104], which were not in evident in *scid* mice and DNA-PKcs^−/−^ mice. In conjunction with this, MEF from Ku80^−/−^ mice and Ku70^−/−^ mice showed reduced proliferation and premature senescence [103,104,105]. Increased cell death in neuronal development was also observed in Ku70^−/−^ mice and Ku80^−/−^ mice [107], although it was less severe than that observed in LIG4^−/−^ mice and XRCC4^−/−^ mice (see next).

LIG4^−/−^ mice and XRCC4^−/−^ mice exhibited late embryonic lethality [108,109,110]. MEF from LIG4^−/−^ mice and XRCC4^−/−^ mice showed reduced proliferation and premature senescence, as well as an increased sensitivity to IR [108,109,110]. Mature B and T lymphocytes were absent in these mice, and the fibroblasts were defective in both coding joint formation and signal joint formation in V(D)J recombination. In addition, the defective neuronal development associated with greatly increased cell death was manifested in LIG4^−/−^ mice and XRCC4^−/−^ mice [109,110].

Artemis^−/−^ mice grew normally but were deficient in lymphocyte development [111]. Artemis^−/−^ MEF showed increased IR sensitivity [111]. While coding joint formation in Artemis^−/−^ MEF was greatly impaired, signal joint formation was indistinguishable from the wild-type in rate and fidelity [111]. Since DNA-PKcs^−/−^ MEF showed mildly reduced fidelity, DNA-PKcs might have an Artemis-independent function in signal joint formation.

XLF^−/−^ mice did not exhibit overt defects in growth and development [112]. There was a slight decrease in the number of lymphocytes, but the distribution of mature lymphocytes was normal, although a mild defect in the class switch recombination (CSR) was evident [112]. However, ES cells or MEFs showed increased IR sensitivity and V(D)J recombination defects in both signal joint formation and coding joint formation [112]. These results suggested the presence of a lymphocyte-specific mechanism to compensate for a XLF deficiency. XLF is shown to have functional redundancy with ATM [113], DNA-PKcs [114] and PAXX (see next). Of note, DNA-PKcs^−/−^; XLF^−/−^ mice showed a reduction in birth ratio and body size at birth [114]. Additionally, signal joint formation in V(D)J recombination was compromised in DNA-PKcs^−/−^; XLF^−/−^ mice, although it was mostly normal in Artemis^−/−^, XLF^−/−^ mice [113,114]. This observation further supports the Artemis-independent function of DNA-PKcs in signal joint formation.

PAXX^−/−^ mice also showed normal growth and development, although they showed a slightly reduced survival after γ-ray irradiation [115,116]. PAXX^−/−^; XLF^−/−^ mice were embryonic lethal, dying between E14.5 and E18.5. A reduced body size became evident at around E10.5 [115,116], suggesting possible redundant functions between PAXX and XLF.

These lines of evidence indicate that NHEJ is essential in growth and development. We can also see that the requirement for DNA-PKcs is less pronounced than that of Ku70, Ku80, XRCC4 and LIG4.

### 2.4. Human Patient and Cells Deficient in DNA-PKcs: Manifested Importance in Human

To date, six human individuals have been reported to harbor homozygous or compound heterozygous mutations in DNA-PKcs, as shown in Table 5. Five patients (P1 and P3-P5) are of Turkish origin and have a common homozygotic mutation. Of two mutations, i.e., one deletion and one substitution of amino acid in each allele, the latter is considered responsible for the disease [117]. It is noteworthy that the expression of DNA-PKcs, its autophosphorylation on Ser2056 and kinase activity appeared normal in the fibroblast derived from P1 [117]. Moreover, when exogenously expressed in V3 cells, the mutated gene could accumulate on DNA damage induced by laser micro-irradiation and recruit Artemis there as well [117]. These lines of evidence indicate that this mutation is hypomorphic, retaining a substantial part of the DNA-PKcs functions. Nevertheless, this mutation increased the length of the P-nucleotide in the coding joint, as in the case of patients with mutations in Artemis, indicating that this mutant might be defective in the activation of Artemis [117]. Other patients with the same mutations showed granuloma and/or autoimmunity, as well as SCID [118,119].

One patient (P2) of British origin had a distinct compound heterozygotic mutation [120]. This patient was first given clinical attention due to poor intrauterine growth and, after birth, showed various symptoms, including microcephaly, facial dysmorphism, seizures and other neurological abnormalities, in addition to SCID [120]. He died at 31 months of age because of intractable seizures [120]. Unlike the case of P1, the expression of DNA-PKcs in the fibroblast from P2 was very low, and the kinase activity was not detectable [120]. These observations suggested a more severe defect in DNA-PKcs functions in P2 than in P1, causing growth and neuronal defects in addition to immunodeficiency. Nonetheless, the DNA-PKcs in this patient might have been partially functional, because the treatment of fibroblasts from the patient prolonged the decline of the γ-H2AX foci after IR [120]. Additionally, the A3574V mutant could partially restore the coding joint proficiency, with normal sequence, to DNA-PKcs-deficient V3 cells, suggesting that this mutant is capable of activating Artemis [120].

Thus, human individuals with null-functional DNA-PKcs have not been found so far and might not be viable at all. In addition, the attempts to establish Ku70^−/−^ and Ku80^−/−^ cells from HCT116 or Nalm-6 failed, and these genes proved indispensable for the viability of human cells [121,122,123]. The lethality might be due to telomere deletion in the form of telomeric circles, which should be suppressed by Ku86 [123]. Ku86 was also shown to suppress alternative, DNA polymerase θ-mediated NHEJ (A-NHEJ, A-EJ or TMEJ), which are thought more susceptible to errors than classical NHEJ [124]. The expression of DNA-PKcs and DNA-PK kinase activity in human cells were reported to be higher than in rodent cells by one or two orders of magnitude [11,17,96]. Human colon cancer HCT116 cells exhibit a haploinsufficiency of DNA-PKcs in terms of various functions; DNA-PKcs^+/-^ showed a slower proliferation and higher sensitivity to IR and etoposide, as well as a shorter telomere length than DNA-PKcs^+/+^ cells [77] (Table 1). Ku70 and Ku80 also exhibit haploinsufficiency in terms of cell proliferation, IR sensitivity and telomere length [121,122,123,124,125]. On the other hand, cells lacking XRCC4 or LIG4 were established from several cell lines and were shown to be viable [78,79,126,127,128,129,130]. This is in contrast to the situation in mice, where the absence of XRCC4 and LIG4 results in more severe consequences than the absence of Ku70 or Ku80. These lines of evidence indicate that the importance of DNA-PKcs and Ku might be manifested in humans as compared to other mammalian species.

## 3. Relationship between DNA-PKcs and ATM

### 3.1. Overlap and Nonoverlap in Functions between DNA-PKcs and ATM

ATM^−/−^ mice showed growth retardation, neurological dysfunction, immunodeficiency, infertility and increased tumor susceptibility, as observed in human AT patients [131,132]. As the mice doubly deficient for ATM and DNA-PKcs, Ku80 or Ku70 showed early embryonic lethality (E11.5–13.5) [133,134]. This is substantially different in timing from the late embryonic lethality (after E13.5) observed in LIG4^−/−^ or XRCC4^−/−^ mice. Cultured cells deficient in ATM and DNA-PKcs have not been established to date and are thought to be inviable. Therefore, ATM and DNA-PKcs are thought to have partially overlapping roles required to sustain cell survival and animal development.

Eriquez-Rios et al. [135] examined the interrelationship of DNA-PKcs, ATM and ATR in the brains of mice embryos using DNA-PKcs^−/−^ mice, neuron-specific ATM/ATR knockout mice (Atm^Nes-Cre^ and Atr^Nes-Cre^, respectively) and their crosses. Although DNA-PKcs^−/−^ mice have shown normal neural development, increased DNA damage and apoptosis were observed after irradiation, especially in postmitotic cells [135]. This observation indicated the role of DNA-PKcs in protecting cells from DNA damage and apoptosis [135]. This function might be manifested in human, and its impairment might lead to neuronal abnormalities, as observed in one of the reported patients (P2) [135]. The radiation-induced apoptosis was less pronounced in DNA-PKcs^−/−^; Atm^Nes-Cre^ than DNA-PKcs^−/−^, indicating that ATM regulates apoptosis [135]. Atr^Nes-Cre^ mice showed increased M-phase cells in proliferating cells, indicating a defective G2/M checkpoint [135]. Thus, in developing neuronal systems, DNA-PKcs, ATM and ATR seem to play major roles in DNA repair, apoptosis and the cell cycle checkpoint, respectively.

A fibroblast cell line from an AT patient was shown to be proficient in the signal joint formation and coding joint formation of extrachromosomal plasmid substrates [136]. However, the coding joint formation of the chromosome-integrated substrate was reduced in pre-B cells from ATM^−/−^ mice, as compared to ATM^+/+^ mice [137]. ATM^−/−^ mice also exhibited impairment in the maturation of B cells and T cells due to defective coding joint formation in immunoglobulin and TCRα [138,139]. On the other hand, the signal joint formation appeared mostly normal in ATM^−/−^ mice [113]. Gapud et al. [140] and Zha et al. [141] demonstrated that the treatment of DNA-PKcs^−/−^ cells and ATM^−/−^ cells with an ATM inhibitor (ATMi) and DNA-PK inhibitor (DNA-PKi), respectively, resulted in impairment of the signal joint formation. Zha et al. also showed similar results by the conditional knockout of the DNA-PKcs gene in the ATM^−/−^ background [141]. These results indicated that DNA-PKcs and ATM have redundant roles in signal joint formation. As described above, the kinase activity of DNA-PKcs is required for V(D)J recombination, as well as the repair of radiation-induced DSBs. Since DNA-PKcs and ATM share similar biochemical properties, there can be redundancy between these kinases in the phosphorylation of protein(s), which is discussed next.

### 3.2. Overlap and Nonoverlap in Protein Phophorylation by DNA-PKcs and ATM in NHEJ

Artemis is shown to be capable of opening the hairpin end and processing the overhang in a manner dependent on DNA-PKcs and its kinase activity [51]. Where Artemis was shown to be phosphorylated by DNA-PK at 11 sites in vitro [142,143], the mutants lacking these phosphorylation sites did not show a reduction in the enzymatic activities and V(D)J recombination functions [142,143,144]. It was shown that the autophosphorylation of DNA-PKcs was required for Artemis activation [144]. IR-induced Artemis phosphorylation in cellulo was shown to be mediated by ATM, as well as DNA-PKcs [142,143,144,145,146,147]. It was shown later that DNA-PK kinase activity was dispensable for coding joint formation when three phosphorylation sites by ATM were present and the ATM activity was not perturbed [148]. This observation suggests that, at least one of the redundant function(s) of DNA-PKcs and ATM in coding joint formation is to phosphorylate DNA-PKcs itself. In addition to coding joint formation, the interaction of Artemis and ATM was implicated in the cell cycle checkpoint [145] and HR [146].

XRCC4 was shown to be phosphorylated by DNA-PK in vitro [46,149,150] and in cellulo in response to IR [151]. Ser260 and Ser320 (Ser318 in an alternatively spliced form) were identified as the major phosphorylation sites in vitro [152,153]. The phosphorylation of these serines in cellulo in response to IR is mostly diminished by treatment with a DNA-PK inhibitor or in DNA-PKcs-deficient cells [154,155]. On the other hand, the ATM inhibitor alone did not show a discernable reduction in phosphorylation [154,155], indicating that XRCC4 phosphorylation at these sites is mostly mediated through DNA-PK. However, the XRCC4 mutants lacking these phosphorylation sites retained normal functions in in the restoration of radioresistance and V(D)J recombination in XRCC4-deficient cells and, also, in the DNA joining reaction in the cell-free system [152,153], although it was also reported that the mutant lacking Ser260 showed a slight but significant increase in radiosensitivity and decrease in DSB repair ability [155]. There are additional phosphorylation sites in XRCC4 by DNA-PK, and further studies are required to clarify the significance of phosphorylation. Normanno et al. [156] replaced eight potential phosphorylation sites (Ser193, Ser260, Ser304, Ser315, Ser320, Thr323, Ser327 and Ser328) in XRCC4 with alanine or aspartate and showed that none of them could fully restore the radioresistance of the XRCC4-deficient cells. It was also shown that the aspartate-substituted (phospho-mimic) mutant showed a decreased DNA bridging ability and increased dissociation of the XRCC4-XLF complex in DNA [156]. It may be noted that seven of the potential phosphorylation sites, i.e., except for Ser193, are located in the intrinsically disordered C-terminal region of XRCC4 and that four of them are clustered in the XRCC4 extremely C-terminal (XECT) region, which is unique to and highly conserved among vertebrates [157].

XLF was also shown to be phosphorylated by DNA-PK in vitro, and Ser245 and Ser251 were identified as the major phosphorylation sites [158]. In cellulo, Ser245 is phosphorylated by DNA-PK, whereas Ser251 is phosphorylated by ATM [158]. However, the XLF mutant lacking these phosphorylation sites could restore the radioresistance and DSB repair ability to XLF-deficient cells, indicating that the phosphorylation of XLF at these sites is dispensable for DNA repair function [158]. As in the case of XRCC4, Normanno et al. [156] generated mutants, in which six potential phosphorylation sites (Ser132, Ser203, Ser245, Ser251, Ser263 and Ser266) were substituted with alanine and aspartate, and showed that none of them could fully restore the radioresistance of XLF-deficient cells [156]. The aspartate-substituted (phospho-mimic) mutant of XLF showed a reduced DNA bridging ability and increased the dissociation of the XRCC4-XLF complex from DNA [156].

The accumulated evidence indicates the main function of DNA-PK in NHEJ and that of ATM in HR and the cell cycle checkpoints. As seen above, the role of the protein phosphorylation by DNA-PK in NHEJ is still largely unclear, demanding further studies. DNA-PK is also shown to phosphorylate a great number of proteins in vitro and in vivo, which are not currently implicated in NHEJ (as reviewed in reference [20]). The first reported substrate of ATM was p53 [37,88]. Thereafter, ATM was shown to phosphorylate a number of proteins in vitro and in vivo, including NBS1 [159,160,161] and BRCA1 [162,163], which are implicated in HR and the cell cycle checkpoints. ATM and DNA-PKcs have a redundant function in the phosphorylation of H2AX on Ser139 (γ-H2AX), where ATM plays a major role [164,165]. γ-H2AX is thought to serve as the “landmark” of DSB. The mediator of DNA damage checkpoint protein 1 (MDC1) binds first to γ-H2AX and, in turn, recruits a series of proteins involved in DSB repair and the cell cycle checkpoint through direct or indirect protein–protein interactions [166]. Matsuoka et al. identified more than 900 phosphorylation sites on more than 700 proteins, which were phosphorylated by ATM or ATR in cellulo, through stable amino acid-labeling in the cell culture (SILAC) combined with mass spectrometry [167]. This evidence suggests extensive networks of DNA damage responses and versatile functions of DNA-PKcs, ATM and ATR therein.

### 3.3. Regulation of DNA-PKcs and ATM by Phosphorylation

In the earliest studies of DNA-PK, it was noticed that DNA-PKcs could phosphorylate itself, i.e., autophosphorylation [2,3]. Since the first identification of autophosphorylation site Thr2609 in 2002 [168], more than 40 serines/threonines have been shown to undergo autophosphorylation in vitro [169,170,171,172,173,174]. There are two major clusters of phosphorylation sites, i.e., the ABCDE cluster around Thr2609 [169,170] and the PQR cluster around Ser2056 [175,176]. Many of them are phosphorylated in response to IR or DNA-damaging agents and essential for DNA repair function [168,169,170,171,172,173,174]. Where the phosphorylation sites in the PQR clusters seems to be autophosphorylated [177], those in the ABCDE cluster are likely phosphorylated by ATR or ATR rather than DNA-PKcs [178]. The phosphorylation in the ABCDE cluster was shown to be essential for the stimulation of Artemis endonuclease activity [148].

It was demonstrated that the autophosphorylation of DNA-PKcs or substitution of serines or threonines in the ABCDE cluster with aspartate to mimic phosphorylation facilitate NHEJ, possibly through changing the conformation of DNA-PKcs itself and increasing the accessibility of repair enzymes to DNA ends [177,179]. Phosphorylation in the ABCDE cluster is also required for the pathway choice between NHEJ and HR. The mutant DNA-PKcs lacking ABCDE phosphorylation sites behaved “even worse” than the functionally null DNA-PKcs. When the mutant was introduced into V3, the transfectant showed higher radiosensitivity than the empty vector transfectant [172,173,174]. As mentioned earlier, the mice lacking three of these phosphorylation sites (DNA-PKcs^3A/3A^) died shortly after birth due to severe bone marrow failure [90]. DNA-PKcs^3A/3A^ MEF showed hypersensitivity to DNA crosslinking agents, like Fanconi’s Anemia (FA) cells, and defects in the HR and FA pathways [90]. The disruption of ABCDE phosphorylation sites might result in the persistence of DNA-PKcs on DSB, preventing it from the HR or FA pathways.

The autophosphorylation of ATM was first shown using the proteins expressed in baculovirus [180]. In 2003, Bakkenist and Kastan showed that ATM undergoes autophosphorylation at Ser1981, which is required for ATM activation through dissociation of the dimer [181]. Since then, the phosphorylation status of ATM Ser1981 has been used extensively as a marker for ATM activation and the cellular response to DSB. The phosphorylation of ATM Ser1981 was also induced after UV irradiation in a manner dependent on ATR rather than ATM [182]. Kozlov et al. identified several additional autophosphorylation sites, including Ser367, Ser1893 [183], Thr1985 and Ser2996 [184], and demonstrated the phosphorylation in cellulo in response to radiation in a manner dependent on ATM and Nbs1. The serines corresponding to Ser1981, Ser367 and Ser1893 in mice were shown to be dispensable for ATM activation [185,186]. On the other hand, the autophosphorylation of Ser1981 was shown to be required for the sustained retention of ATM at DNA-damaged sites [187]. Ser367- or Ser2996-phosphorylated ATM accumulated in heavy ion-induced DNA damage, although they were not required for the accumulation [184]. These serines were required for the S-phase checkpoint function (i.e., to correct the radioresistant DNA synthesis of cells in AT patients) [184].

In 2017, Zhou et al. showed that DNA-PK phosphorylated ATM and MRN in vitro [188]. The phosphorylation of ATM by DNA-PK in vitro inhibited ATM kinase activity [188]. Changing the serines and threonines in two clusters (S85/T86, T372/T373 and T1985/S1987/S1988) into glutamate, mimicking phosphorylation, abolished the ATM activation by MRN and DNA [188]. These mutants showed a deficiency in ATM functions [188]. On the other hand, some the mutants blocking this phosphorylation showed a resistance to the inhibitory effects by DNA-PK. Their study demonstrated that DNA-PK negatively regulates the ATM function through phosphorylation.

Thus, there is accumulating evidence indicating mutual regulation between DNA-PK and ATM through phosphorylation. As there are many potential sites for autophosphorylation and mutual phosphorylation in DNA-PKcs and ATM, further studies are required.

## 4. Conclusive Remarks

DNA-PK, comprised of DNA-PKcs and Ku, acts as the DSB sensor, which is essential for the repair of DSB through NHEJ. As compared to the other factors in NHEJ, DNA-PKcs is especially important for the ligation of “difficult” or “dirty” ends, which need processing before ligation. However, the precise role of the protein phosphorylation in NHEJ remains a great missing link. In addition, the characteristics of animals and human individuals with defective NHEJ show its importance in development, which is manifested in the neuronal and immune systems. DNA-PK has pleiotropic functions beyond DSB repair, which are reviewed elsewhere [20,189]. DNA-PK, as well as other NHEJ factors, are also implicated in carcinogenesis and considered promising targets for cancer therapeutics. There are also many reviews of these aspects, including reference [190].

DNA-PKcs share a similarity in the primary structure and properties with ATM, which is also implicated in cellular responses to DSBs. Compared to DNA-PKcs, ATM shows more pronounced importance in the cell cycle checkpoint. Nevertheless, there are substantial overlaps in the function and protein phosphorylation between DNA-PKcs and ATM. It is also important that DNA-PKcs and ATM undergo extensive and complex regulations through autophosphorylation and mutual phosphorylation.

Since the first identification of DNA-PK activity in animal cell extracts, studies over 35 years have greatly promoted our understanding of the functions and regulatory mechanisms of DNA-PKcs and ATM. However, considering the enormous sizes of these molecules and vast number of substrate proteins and phosphorylation sites therein, our current knowledge might be just the tip of the iceberg, warranting further studies. The outcomes of this research will surely contribute to the etiology and therapeutics of cancer and developmental diseases.

## Figures and Tables

**Figure 1 genes-12-01143-f001:**
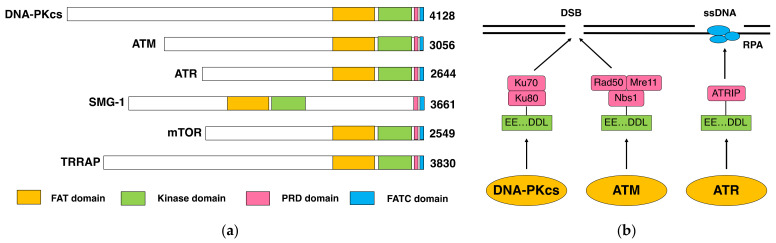
The similarity of DNA-PKcs to ATM and other PIKKs. (**a**) The architecture of DNA-PKcs, ATM and other PIKKs. FAT: FRAP-ATM-TRRAP, PRD: PIKK-regulatory and FATC: FAT C-terminal. (**b**) The mechanisms of the recruitment of DNA-PKcs, ATM and ATR to DSB and ssDNA.

**Figure 2 genes-12-01143-f002:**
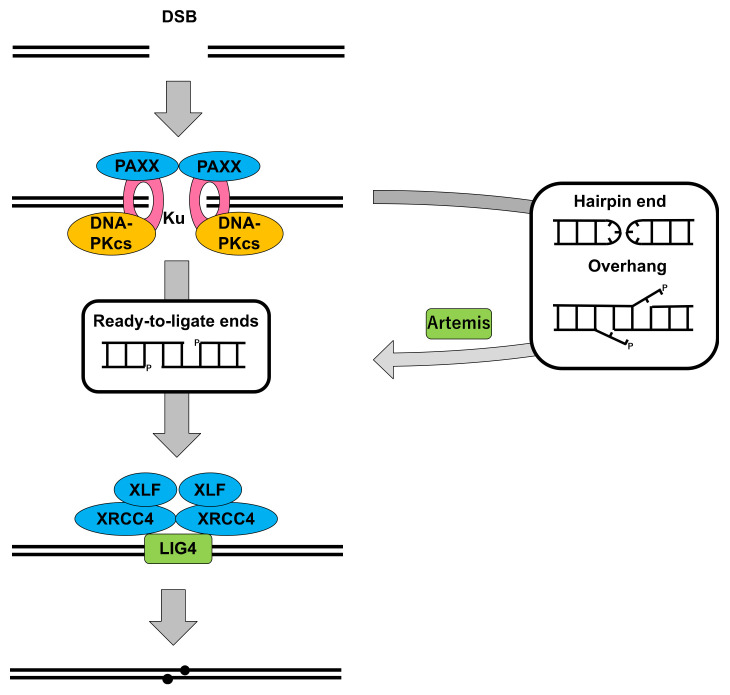
Simplified diagram of DSB repair through NHEJ.

**Table 1 genes-12-01143-t001:** DNA-PKcs-deficient cell lines.

Cell	Species	Type	Mutation ^1^	DNA-PKcs Status	Characteristics	Ref.
*scid*	Mouse	Fibroblast	HMZ c.T12138A, p.Y4046X. (4128 aa)	Protein very low (~1%); DNA binding undetectable; Kinase activity undetectable.	Increased IR-sensitivity; Reduced DSB repair ability; Defective V(D)J recombination.	[10,11,12,66,67,68]
V3	Hamster	Ovary	CHTZ; c.C12070A, p.Q4024X; Not known in the other.(4124 aa)	Protein undetectable; DNA binding barely detectable; Kinase activity undetectable.	Increased IR-sensitivity; Defective V(D)J recombination.	[10,69]
M059J	Human	Glioma	LOH; c.A4051del, p.T1351Pfs*8. (4128 aa)	Protein undetectable; DNA binding barely detectable; Kinase activity undetectable.	Increased IR-sensitivity.	[17,70]
Irs-20	Hamster	Ovary	CHTZ c.G12358A, p.E4120K; 2nd allele not expressed.(4124 aa)	Protein reduced (~10% ^2^); DNA binding reduced (~25% ^2^); Kinase activity undetectable.	Increased IR-sensitivity; Defective V(D)J recombination.	[69,71]
SX-9	Mouse	Mammary carcinoma	CHTZ c.T9572C, p.L3191P; Not known in the other. (4120 aa)	Protein reduced (~5% ^2^); DNA binding reduced (~5% ^2^); Kinase activity undetectable.	Increased IR-sensitivity; Defective V(D)J recombination.	[71,72]
XR-C1	Hamster	Ovary	Unknown.	Protein undetectable; Kinase activity undetectable.	Increased IR- and drug (bleomycin and ethyl methane sulfonate) sensitivity; Defective V(D)J recombination.	[73]
XR-C2	Hamster	Ovary	c.G12353A, p.G4118E. (4124 aa)	Protein expression normal; Kinase activity undetectable.	Increased IR- and drug (bleomycin, ethyl methane sulfonate and mitomycin C)-sensitivity; Reduced DSB repair ability; Defective V(D)J recombination.	[74,75]
(Generated by gene targeting or genome editing)						
DT40	Chicken	B lymphocyte	p. 2888–3012. (4133 aa)	Protein undetectable.	Normal proliferation; Increased IR-sensitivity.	[76]
HCT116	Human	Colon cancer	p. 3831–4127. (4128 aa)	Protein undetectable; Kinase activity undetectable.	Reduced proliferation; Increased IR-, drug (etoposide)-sensitivity; Telomere shortening; Increased chromosomal aberrations.	[77]
TK6	Human	B lymphocyte	Part of exon 32 replaced with drug resistance gene.	Protein undetectable.	Increased IR-sensitivity.	[78]
HAP1	Human	Fibroblast-like, near haploid	11 bp deletion in exon 25.	Protein undetectable.	Increased drug (Etoposide)-sensitivity.	[79]
mESC	Mouse	Embryonic stem	c.24del88; p.R9Wfs7*. (4128 aa)	mRNA very low (<1%).	Upregulation of pluripotency genes.	[80]
HeLa	Human	Cervical carcinoma	Targeting exon 36.	Protein undetectable.	Increased IR-sensitivity.	[81]

^1^ HMZ: homozygote, CHTZ: compound heterozygote and LOH: loss of heterozygosity. Numbers in parentheses indicate the number of amino acids of the full-length DNA-PKcs in the respective species. ^2^ Estimated from the intensity of the Western blotting bands in the references.

**Table 2 genes-12-01143-t002:** V(D)J recombination defects in DNA-PKcs-deficient cells.

Cell	Signal Joint		Coding Joint		Ref.
	Frequency	Structure	Frequency	Structure	
Mouse *scid* (SCGR11)	Normal.	Fidelity slightly decreased (~80%).	Significant decrease (~3%).	Larger deletions.	[82]
V3	Mild decrease (~20%).	Fidelity modestly decreased (~50%).	Significant decrease (~1%).	Abnormally large P elements.	[10]
Mouse *scid* (SCID/St)	Substantial decrease (~10%).	Not described.	Significant decrease (~0.1%).	Not described.	[11]
IRS-20	Substantial decrease (~10%).	Fidelity slightly decreased (~75%).	Significant decrease (~3%).	Smaller deletions than *scid* and V3.	[69]
SX-9	Substantial decrease (~10%).	Fidelity profoundly decreased (~10%).	Significant decrease (~3%).	Slightly longer deletions.	[72]
XR-C1	Significant decrease (~2%).	Correct joins absent (0%).	Significant decrease (~2%).	Not described.	[73]
XR-C2	Mild decrease (~30%).	Not described.	Mild decrease (~50%).	Not described.	[74]
xrs6 ( Ku80)	Significant decrease (~5%).	Fidelity profoundly decreased (~15%).	Significant decrease (~1%).	None recovered.	[82]
XR-V15B (-Ku80)	Undetectable (<1%).	Not described.	Undetectable (<1%).	Not described.	[19]
XR-1 (-XRCC4)	Significant decrease (~2%).	Fidelity profoundly decreased (~20%).	Significant decrease (~0.2%).	Larger deletions.	[45,82]

**Table 3 genes-12-01143-t003:** DNA-PKcs-deficient animals.

Animal	Mutation	DNA-PKcs Status	Animal Phenotype	Cellular Phenotype ^1^	Ref.
Mouse *scid*	c.T12138A, p.Y4046X. (4128 aa)	Protein very low (~1%); DNA binding undetectable; Kinase activity undetectable.	SCID; Increased thymic lymphomas.	Increased IR-sensitivity; Reduced DSB repair ability; Defective V(D)J recombination (CJ but not SJ).	[10,11,12,66,67,68]
Mouse, gene knockout	Insertion of drug resistance gene in exon 6.	mRNA undetectable; Protein undetectable; Kinase activity undetectable.	SCID.	Increased IR-sensitivity.	[84]
Mouse, gene knockout	p. 3860–3950. (4128 aa)	Protein undetectable; Kinase activity undetectable.	SCID.	Defective V(D)J recombination (CJ but not SJ); Increased IR-sensitivity (fibroblast); Normal IR-sensitivity (ES).	[85]
Mouse, gene knockout	3′-half of exon 3 replaced with drug resistance gene.	Protein undetectable.	SCID.	Increased IR-sensitivity.	[86]
Mouse *slip*	A transgene inserted by > 20 copies to upstream of three exons corresponding to 777–1010 nucleotides of mRNA.	mRNA undetectable; Kinase activity undetectable.	SCID; Increased thymic lymphomas.	Not described.	[87,88]
Mouse, KD	c.A11765C, p.D3922A. (4128 aa)	Protein expression normal; Kinase activity undetectable.	Embryonic lethal (E14.5); Defective neuronal development.	Increased IR-sensitivity; Increased genomic instability; Defective V(D)J recombination (CJ and SJ).	[89]
Mouse, 3A	c.A7813/7900 /7927G p.T2605/2634/2643A. (4128 aa)	Protein expression normal; Kinase activity normal.	Born at normal ratio and size, but becomes smaller 2–3 weeks of age; Death shortly after birth (75% within 4 w); Congenital bone marrow failure; Loss of hematopoietic stem cells.	Increased sensitivity to IR, UV, CPT and MMC.	[90]
Mouse, Balb/c, C.B.17, 129	c.C6418T, p.R2140C/c.A11530G, p.M3844V.	Protein expression decreased (5–10%); Kinase activity reduced (5–10%).	Immunologically normal; Normal development; Increased thymocyte apoptosis; Susceptible to cancer, including breast cancer and thymic lymphoma.	Delay in DSB repair; Increased chromosome instability.	[91,92,93]
Horse SCID (Arabian foal)	c.9478del5, p.S3160Nfs4*. (4134 aa)	Protein undetectable; Kinase activity undetectable.	SCID.	Increased IR-sensitivity; Defective V(D)J recombination (CJ and SJ).	[94,95,96]
Dog SCID (Jack Russel Terriers)	c.G10879A, p.E3627X. (4144 aa)	Protein undetectable; DNA binding barely detectable; Kinase activity undetectable.	SCID.	Increased IR-sensitivity; Defective V(D)J recombination (CJ and SJ).	[97,98]
Rat, gene knockout	Deletion in exon 1, causing frame-shift. (4126 aa)	mRNA undetectable; Protein undetectable.	SCID; Defective lymphocyte development; Growth retardation; Reduced litter size (~1/2).	Reduced proliferation; Premature senescence; IR-sensitivity; Defective NHEJ.	[99]
Zebrafish, gene knockout	Frame-shift in exon 3. (4119 aa)	Protein undetectable.	SCID; Growth delay up to 3 months.	Not described.	[100]
Zebrafish, gene knockout	c.10835del8, p.D3612Vfs17. (4119 aa)	Not described.	SCID; IR-sensitivity.	Not described.	[101]

^1^ CJ: coding joint and SJ: signal joint.

**Table 4 genes-12-01143-t004:** Gene knockout mice of DNA-PKcs and other NHEJ genes.

Mice	Viability	Growth	Neurogenesis	Immunity ^1^	Ref.
DNA-PKcs^−/−^	Viable.	Normal body size.	Normal.	SCID (leaky); SJ: normal or modestly impaired; CJ: impaired.	[84,85,86,107]
Ku80^−/−^	Viable.	Reduced body size.	Defective (milder than XRCC4^−/−^ and LIG4^−/−^); Increased cell death.	SCID; SJ & CJ: defective.	[102,103,107]
Ku70^−/−^	Viable.	Reduced body size.	Defective (milder than XRCC4^−/−^ and LIG4^−/−^); Increased cell death.	SCID (leaky); SJ & CJ: defective.	[104,105,107]
LIG4^−/−^	Late embryonic lethality (>E13.5)	Reduced body size in uterus.	Severely defective; Massive cell death.	SCID; SJ & CJ: defective.	[108,109]
XRCC4^−/−^	Late embryonic lethality (>E13.5)	Reduced body size in uterus.	Severely defective; Massive cell death.	SCID; SJ & CJ: defective.	[110]
Artemis^−/−^	Viable.	Normal body size.	Normal.	SCID (leaky); SJ: normal; CJ: impaired.	[111]
XLF^−/−^	Viable.	Normal body size.	Normal.	Mostly normal; Slight decrease in the number of lymphocytes; Normal lymphocyte distribution; Mild defect in CSR.	[112]
PAXX^−/−^	Viable.	Normal body size.	Normal.	Mostly normal; Modest decrease in the number of lymphocytes.	[115,116]

^1^ SJ: signal joint formation and CJ: coding joint formation.

**Table 5 genes-12-01143-t005:** Human patients deficient in DNA-PKcs.

Patient	Gender	Ethnic Origin	Mutation ^1^ and DNA-PK Status	Clinical Characteristics	Cellular Characteristics	Ref.
P1 (ID177)	F	Turkish	HMZ c.6338del3, p.G2113del/ c.T9185G, p.L3062R. Protein expression normal; Kinase activity normal.	SCID.	Increased IR-sensitivity; Delay in DSB repair.	[117]
P2 (NM720)	M	British	CHTZ; c. exon16; c.C10721T, p.A3574V. Protein very low (~5% ^2^); Kinase activity undetectable.	SCID; Growth failure; Microcephaly; Facial dysmorphism; Seizures; Bilateral sensorineural hearing loss; Visual impairment; Died at 31 months.	Increased IR-sensitivity; Delay in DSB repair.	[120]
P3	M	Turkish	HMZ c.6338del3, p.G2113del/ c.T9185G, p.L3062R.	SCID; Granuloma; Autoimmunity.	Not described.	[118]
P4	F	Turkish	HMZ c.6338del3, p.G2113del/ c.T9185G, p.L3062R.	SCID; Granuloma; Autoimmunity.	Not described.	[118]
P5	F	Turkish	HMZ c.6338del3, p.G2113del/ c.T9185G, p.L3062R.	SCID; Granuloma; Arthritis.	Not described.	[119]
P6	F	Turkish	HMZ c.6338del3, p.G2113del/ c.T9185G, p.L3062R.	SCID; Granuloma; Diarrhea.	Not described.	[119]

^1^ HMZ: homozygote and CHTZ: compound heterozygote. ^2^ Estimated from the intensity of the Western blotting bands in the references.

## Data Availability

Not applicable.

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
