# Peer review of "DNA-Dependent Protein Kinase Catalytic Subunit: The Sensor for DNA Double-Strand Breaks Structurally and Functionally Related to Ataxia Telangiectasia Mutated"

_genes, 2021, doi:10.3390/genes12081143_

Round 1

Reviewer 1 Report

Overall, I think this is an interesting and important review paper.  I feel Figure 2 is a bit oversimplified, but is acceptable as this is not a main topic of this ms.  

Author Response

We greatly appreciate your comments on our review.

We agree that Figure 2 is oversimplified. We intended to show the relationships among NHEJ factors discussed here. To clarify this intention, we changed the title of the figure to “Simplified diagram of DSB repair through NHEJ” and added a few sentences in the text (section 2.1, lines 739-742).

Reviewer 2 Report

This review paper summarized the role of DNA-PK on DNA double strand breaks (DSB) repair as the sensor of DSB. Overall, the manuscript is well written and well organized. However, new insight on the role of DNA-PK was not described. This paper is like a textbook.

Authors may discuss about the role of DNA-PK beyond DNA repair or about targeting DNA-PK for cancer treatment as reported in other papers (DOI:10.1158/2159-8290.CD-14-0358, https://doi.org/10.1016/j.mrfmmm.2020.111692, https://doi.org/10.3389/fonc.2019.00635 ).

Major issue

Authors can delete the 1st section (Historical perspectives), because authors have already introduced this in a previous paper (Ref 86).

Minor issue

Correct some errors. line 77 that that DNA-PK, line 78 dsDNA DNA

Author Response

Answer to overall comments.

We greatly appreciate your comments on our review. Since the topic of this special issue is “Role of ATM and MRE11 in Genomic Stability and Oxidative Stress Responses”, we have written this review focusing on DNA-PK in DNA double-strand break repair and its relationship to ATM. We strongly agree that there are other interesting and cutting-edge topics like functions beyond DNA repair and application in cancer therapy. As these issues will be too big to be discussed here, we have added brief comments in the Concluding Remarks (lines 553-556), citing two of the references following your suggestion.

Answer to major issue

We agree that section 1 have redundancy with our recent review. Therefore, we condensed these parts, providing minimal information for understanding the topic in this review: what are DNA-PK and DNA-PKcs and what is the relationship between DNA-PKcs an ATM. Especially, the section 1.1 and 1.2 are combined and reduced to less than half (85 lines to 41 lines).

Some information regarding V(D)J recombination and complementation groups was moved to section 2.1 (lines 149-157).

As a result, we could shorten our manuscript by two pages.

Answer to minor issue

Thank you for pointing out our mistakes. We have corrected them accordingly.